# Genome editing to model and reverse a prevalent mutation associated with myeloproliferative neoplasms

**Ron Baik[1,2,3], Stacia K. Wyman[1,2], Shaheen Kabir[1,2,4]\*, Jacob E. Corn[1,2¤]\***

**1** Innovative Genomics Institute, University of California, Berkeley, CA, United States of America, **2** Department of Molecular and Cell Biology, University of California, Berkeley, CA, United States of America, **3** New York University School of Medicine, New York, NY, United States of America, **4** Helen Diller Family Comprehensive Cancer Center, University of California, San Francisco, CA, United States of America

¤ Current address: Department of Biology, ETH Zurich, Zurich, Switzerland
\* Jacob.corn@biol.ethz.ch (JEC); shaheen.kabir@ucsf.edu (SK)

**Data Availability Statement:** All relevant data are within the manuscript and its Supporting Information files.

## Abstract

Myeloproliferative neoplasms (MPNs) cause the over-production of blood cells such as erythrocytes (polycythemia vera) or platelets (essential thrombocytosis). JAK2 V617F is the most prevalent somatic mutation in many MPNs, but previous modeling of this mutation in mice relied on transgenic overexpression and resulted in diverse phenotypes that were in some cases attributed to expression level. CRISPR-Cas9 engineering offers new possibilities to model and potentially cure genetically encoded disorders via precise modification of the endogenous locus in primary cells. Here we develop "scarless" Cas9-based reagents to create and reverse the JAK2 V617F mutation in an immortalized human erythroid progenitor cell line (HUDEP-2), CD34+ adult human hematopoietic stem and progenitor cells (HSPCs), and immunophenotypic long-term hematopoietic stem cells (LT-HSCs). We find no overt *in vitro* increase in proliferation associated with an endogenous JAK2 V617F allele, but co-culture with wild type cells unmasks a competitive growth advantage provided by the mutation. Acquisition of the V617F allele also promotes terminal differentiation of erythroid progenitors, even in the absence of hematopoietic cytokine signaling. Taken together, these data are consistent with the gradually progressive manifestation of MPNs and reveals that endogenously acquired JAK2 V617F mutations may yield more subtle phenotypes as compared to transgenic overexpression models.

## Introduction

The discovery of programmable endonucleases has dramatically altered our ability to manipulate human genomes. The simplicity and robustness of CRISPR-Cas9 mediated genome engineering allows for significant advances in modeling and treating genetic disease, especially in cells that can both self-renew and differentiate, such as human hematopoietic stem cells (HSCs). Precise genetic manipulations in HSCs provide a powerful research tool to investigate the mechanisms of germline and somatic genetic blood disorders and could revolutionize the

**Funding:** This work was supported by the California Institute of Regenerative Medicine (DISC1-08776), the Li Ka Shing Foundation, the Heritage Medical Research Institute, and the National Institutes of Health New Innovator Awards (DP2 HL141006).

**Competing interests:** The authors would also like to declare that no competing interests exist.

treatment of hematological malignancies. Interrogation of gene-function relationships of monogenic hematological disorders is particularly attractive since these disorders are amenable to development of editing therapies targeting a single locus [1–3].

Myeloproliferative Neoplasms (MPNs) are genetic blood disorders characterized by unbridled proliferation of erythroid, myeloid, and/or megakaryocytic lineages. MPNs can lead to thrombohemorrhagic events, vascular complications, splenomegaly, progressive cytopenia and hypercellular bone marrow [4–6]. MPNs normally manifest later in life, with a median age of 60, and are grouped into three categories: polycythemia vera (PV), essential thrombocytosis (ET), and primary myelofibrosis (PMF). PV is primarily characterized by excess production of erythroid lineages, but megakaryocytic and granulocytic expansion can also be observed. ET is characterized by increased platelets in addition to megakaryocytic hyperplasia. PMF displays heterogeneous phenotypes including bone marrow fibrosis, megakaryocytic hyperplasia and eventually splenomegaly [7].

MPNs arise from clonal expansion of somatically mutated HSCs or progenitors. For PV, 95% of patients harbor a G>T substitution in exon 14 of the Janus Kinase 2 (*JAK2*) gene, causing a valine to phenylalanine mutation at residue 617 (JAK2 V617F). Approximately 60% of patients with ET and PMF also have the same JAK2 V617F mutation [8–11]. Currently available treatments of PV and ET are aimed at prevention of thrombosis and supportive care, while treatment for PMF is mostly palliative [12, 13]. MPN patients treated with JAK2 inhibitors show considerable improvements in blood counts and spleen size, however treatment rarely elicits molecular remission [14]. Allogeneic bone marrow transplantation (BMT) is currently the only curative therapy for MPNs but this is accompanied with severe risks involving toxicity of the transplantation regimen, difficulty in finding a cell donor, and the possibility of graft-versus-host disease [15]. Genome editing approaches to ameliorate MPNs via autologous BMT are therefore an attractive prospect.

JAK2 is required for HSC maintenance and hematopoietic differentiation by constitutively binding hematopoietic cytokine receptors that include the erythropoietin (EPO) receptor, the granulocyte colony-stimulating factor (G-CSF) receptor, and the MPL/thrombopoietin (TPO) receptor. Cytokine binding induces JAK2 activation, which in turn elicits downstream signaling by the Signal Transducer and Activator of Transcription (STAT) proteins to regulate transcription of target genes [16–18]. JAK2 possesses a tyrosine kinase domain responsible for initiating JAK/STAT signaling, and a pseudokinase domain involved in autoregulation. The V617F mutation occurs in the pseudokinase domain of JAK2 and is proposed to abolish autoinhibition, leading to hyperactivation and cytokine-independent JAK/STAT signaling [9, 19–21].

Mouse models of MPN have overexpressed the JAK2 V617F allele transgenically or by transplanting retrovirally transduced murine HSCs. While these informative models exhibit some attributes of MPN, the phenotypes are varied. Some models developed mild or severe PV that eventually progressed to myelofibrosis, while others developed ET [22–26]. Some of these discrepancies were attributed to expression levels of the JAK2 V617F transgene, such that low expression was associated with an ET-like phenotype while higher expression was associated with a PV-like phenotype [25]. Conditional knock-in mice expressing either the murine or human mutant protein under control of the endogenous JAK2 promoter also showed phenotypic variability: heterozygous and homozygous mice expressing murine JAK2 V617F developed PV, while mice heterozygous for human JAK2 V617F developed ET-like symptoms [27–30]. Other somatic co-occurring genetic alterations have also been implicated in the progression of MPN, such as loss-of-function *TET2* mutations. However, recent single-cell sequencing methods that performed simultaneous mutation detection and transcriptome analysis showed that an ET patient acquired the JAK2 V617F mutation before a *TET2* mutation, while an MPN

patient exhibiting bone marrow fibrosis a *TET2* mutation before the JAK2 V617F mutation [31, 32]. Ample literature highlights the importance of JAK2 V617F in MPN, however the early phenotypes following acquisition of the mutation to the onset of frank disease remains unclear.

We developed CRISPR-Cas9 reagents to somatically engineer a "scarless" JAK2 V617F mutation at the endogenous locus in disease-relevant human cells, allowing us to study the effects of acquiring the mutation in human hematopoietic cell lineages. We found that the endogenous JAK2 V617F allele does not lead to excess proliferation in isolation, but confers a growth advantage in a mixed culture setting. JAK2 V617F also hyper-stimulates terminal differentiation of human adult HSCs into beta-globin expressing erythroid cells in the absence of cytokine stimulation. We furthermore developed proof-of-concept gene editing reagents to correct the JAK2 V617F mutation that might in future be useful for autologous transplants of edited CD34+ HSPCs, reducing the risks associated with allogeneic stem cell transplants.

## Results

### Generation of JAK2-V617 editing reagents

To endogenously edit the *JAK2* locus via homology directed repair (HDR), we screened a panel of single guide RNAs (sgRNAs) to find those that efficiently induced Cas9-mediated editing proximal to the V617 amino acid. We designed 10 sgRNAs within a 200 bp window centered on JAK2 V617 (**Fig 1A and S1 Fig**) and tested them by nucleofecting individual Cas9-sgRNA ribonucleoprotein (RNP) complexes into K562 human erythroleukemia cells. Three days after electroporation, we performed a T7 Endonuclease I (T7E1) assay on edited cell pools and found most of the sgRNAs elicited robust insertions and deletions (indels) at the target locus (**Fig 1B**). Because increasing distances between Cas9 cut site and donor-encoded mutations can reduce HDR efficiency, we chose to further pursue sgRNA-4 and sgRNA-6 due to their close proximity to the desired edit site [33, 34]. The sgRNA-4 cut site directly abuts position 617 which should favor high-efficiency HDR, but its location within a coding region also introduces the possibility of co-occurring indels that could yield a premature stop codon (**Fig 1A**). SgRNA-6 targets an intronic site 26 bp downstream of the desired mutation site, which could be less desirable for HDR due to the large distance between sgRNA recognition site and the mutation, but short indel alleles within the intron would not be predicted to interfere with the coding sequence.

We designed a "617F" single-stranded oligonucleotide donor (ssODN) encoding the patient-observed G>T base substitution to generate the JAK2 V617F mutation via HDR (**Fig 1A**). This G>T V617F mutation abolishes a naturally occurring BsaXI restriction site (**Fig 1A**), which we later used to screen for successfully edited clones. The V617F mutation lies within the seed region of sgRNA-4, which is predicted to prevent re-cutting upon introduction of the mutation [35, 36]. The location of the JAK2 V617F mutation within the seed region of sgRNA-4 also allowed us to design a mutation-specific sgRNA (F617-sgRNA) that we reasoned might only target mutated cells, despite sharing the same PAM as sgRNA-4 (**Fig 1A**).

We nucleofected Cas9 pre-complexed with either sgRNA-4 or sgRNA-6 into human K562 erythroleukemia cells together with the 617F ssODN. We quantified the frequencies of NHEJ- and HDR-mediated repair by next-generation sequencing of PCR amplicons (amplicon-NGS) derived from genomic DNA extracted from pools of edited cells after 3 days. Editing with sgRNA-4 resulted in significantly higher HDR (13.6 ± 0.5%) than sgRNA-6 (1.9 ± 0.1%) (**Fig 1C**). To determine whether these results were consistent across cell types and in more disease-relevant cells, we performed editing with sgRNA-4 or -6 plus the 617F ssODN in human CD34 + adult mobilized hematopoietic stem/progenitor cells (HSPCs). Indel frequencies were

**a.**

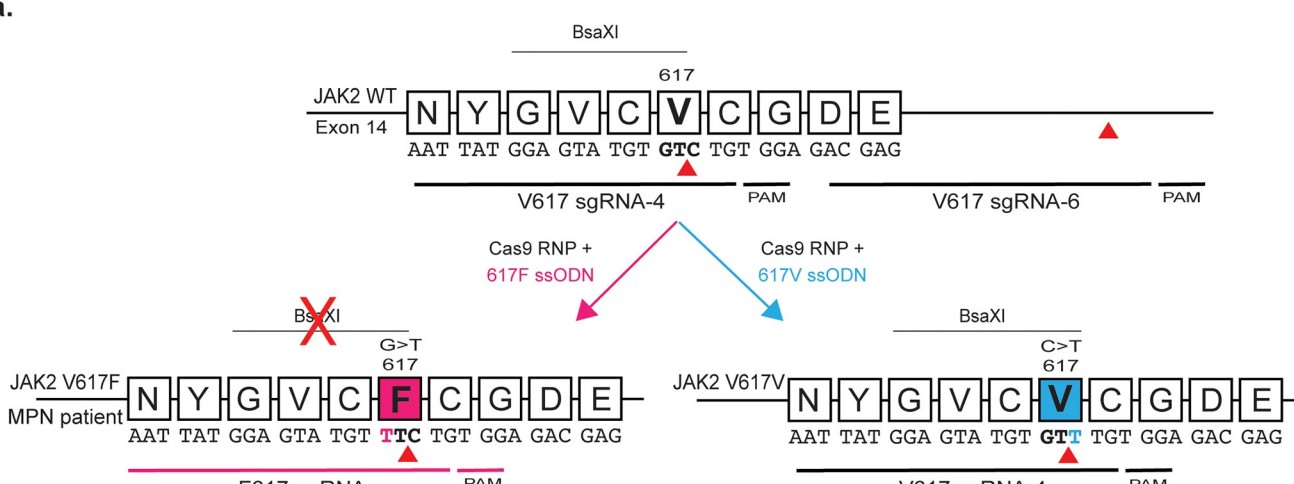

**b.**

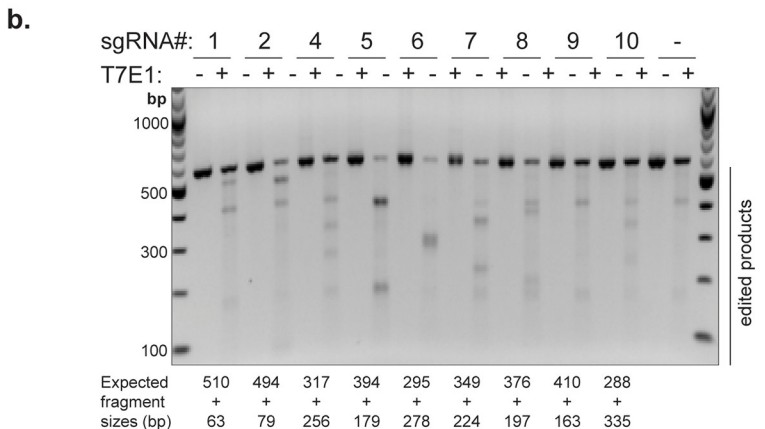

**c.**

**d.**

**e.**

sgRNA-4 + 617F ssODN

sgRNA-4 + 617V ssODN

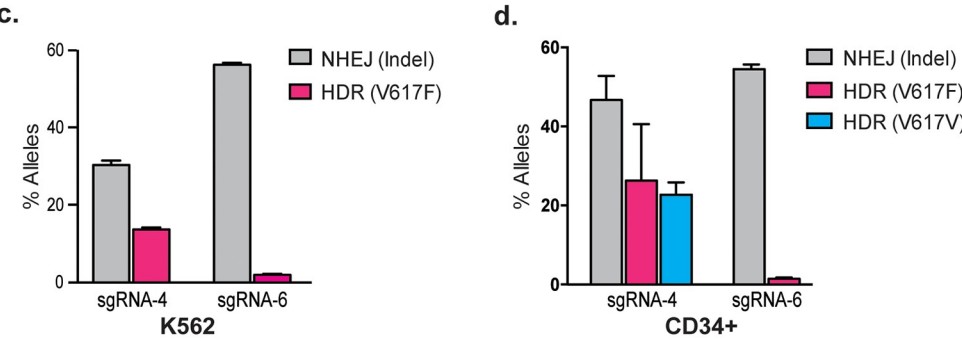

**Fig 1. Cas9-RNP and ssODN-mediated targeting of the JAK2 locus in K562s and CD34+ HSPCs. (a)** Schematic of the JAK2 V617 locus with relevant genetic editing reagents annotated. SgRNAs-4 and -6 are identified with their corresponding 5'-NGG-3' PAMs. Red arrowheads indicate cut site directed by these sgRNAs. The conversion of G>T results in the amino acid change from valine to phenylalanine (V617F, magenta) and removes the naturally occurring BsaXI restriction site. The conversion of C>T results in a silent mutation (V617V, blue) and retains the BsaXI restriction site. **(b)** T7 endonuclease I assay showing indel formation in cell pools edited independently with 9 different sgRNAs. **(c)** K562 cells were nucleofected with Cas9-RNP and the 617F ssODN. NHEJ- and HDR-mediated editing outcomes were assessed by amplicon-NGS. Data from n = 3 independent biological replicates with mean±SD graphed. **(d)** CD34+ HSPCs were nucleofected with Cas9-RNP and 617F or 617V ssODN. NHEJ- and HDR-mediated outcomes were assessed by amplicon-NGS. Data from n≥3 independent biological replicates with mean±SD graphed. **(e)** Allele spectra and corresponding percentages of alleles generated following editing with sgRNA-4 RNP and 617F or 617V ssODN in CD34+ HSPCs. Mutations causing frame-shifts are outlined.

comparable between sgRNA-4 (46.7 ± 6.1%) and sgRNA-6 (54.5 ± 1.2%) in HSPCs, but we consistently observed approximately 26.3 ± 14.3% JAK2 V617F mutation incorporation with sgRNA-4 and only 1.4 ± 0.4% with sgRNA-6 (**Fig 1D**).

We reasoned that modeling *JAK2* mutations would be complicated by the very low levels of HDR stemming from sgRNA-6. Hence, we used sgRNA-4 for subsequent experiments and designed a "silent" HDR strategy to separate the effects of the JAK2 V617F mutation from co-occurring indel alleles or other unanticipated side-effects of editing. A 617V ssODN was created to pair with sgRNA-4 and generate a C>T substitution encoding a synonymous V617V mutation, acting as a "clean" control (**Fig 1A**). This 617V ssODN is also theoretically capable of "correcting" the F617V allele back to V617.

RNP electroporation of CD34+ HSPCs with sgRNA-4 and the 617V ssODN followed by amplicon-NGS revealed HDR frequencies of 22.7 ± 3.2%, similar to the V617F reagents (**Fig 1D**). HSPCs edited with sgRNA-4 and the 617F or 617V ssODNs harbored approximately 47.2 ± 8.5% and 46.3 ± 6.2% indel alleles, respectively. Of these indel alleles, 30.4% (for 617F ssODN) and 36% (for 617V ssODN) were frameshifts (**Fig 1E**). The distribution of different indel alleles was similar when using either 617F or 617V donors. Overall, we developed sgRNAs and ssODNs to engineer high levels of HDR creating several JAK2 mutations: V617F (MPN), V617V (synonymous), and potentially F617V (therapeutic).

## JAK2 V617F introduces a competitive growth advantage in immortalized CD34+ progenitors

To molecularly characterize the effects of acquiring the JAK2 V617F mutation, we utilized a recently established CD34+ immortalized human erythroid progenitor cell line that resembles a primitive hematopoietic progenitor cell population. *H*uman *u*mbilical cord blood-*d*erived *e*rythroid *p*rogenitor (HUDEP-2) cells can undergo terminal erythroid differentiation into erythroblasts with functional hemoglobin and express erythroid-specific cell surface markers such as Glycophorin A (GlyA) and CD71 [37].

We isolated an allelic series of HUDEP-2 clones containing the JAK2 V617F mutation by nucleofecting sgRNA-4 programmed Cas9-RNP together with the 617F ssODN and seeding the edited pool as single cells by fluorescence-activated cell sorting (FACS) (**Fig 2A**). Because wild type HUDEP-2s are a polyclonal mixture, we also isolated multiple wild type clones to control for clonal effects. The G>T V617F mutation abolishes a naturally occurring BsaXI restriction site (**Fig 1A**), which we used to rapidly screen candidate clones (**S2A Fig**). We isolated several wild type clones, two heterozygous JAK2 V617F clones (H1-H2), and three homozygous JAK2 V617F clones (F1-F3). We determined exact genotypes for each clone by TA-cloning and Sanger sequencing (**S2B Fig**).

To test our potentially therapeutic reagents, we nucleofected WT and V617F mutant HUDEP-2 clones with F617-sgRNA and the 617V ssODN. We found no detectable editing in WT clone, confirming that the seed region mutation of F617-sgRNA was sufficient to spare the WT allele. The V617F clone F1 exhibited 62.5 ± 3.5% indels and 20 ± 2.1% HDR back to

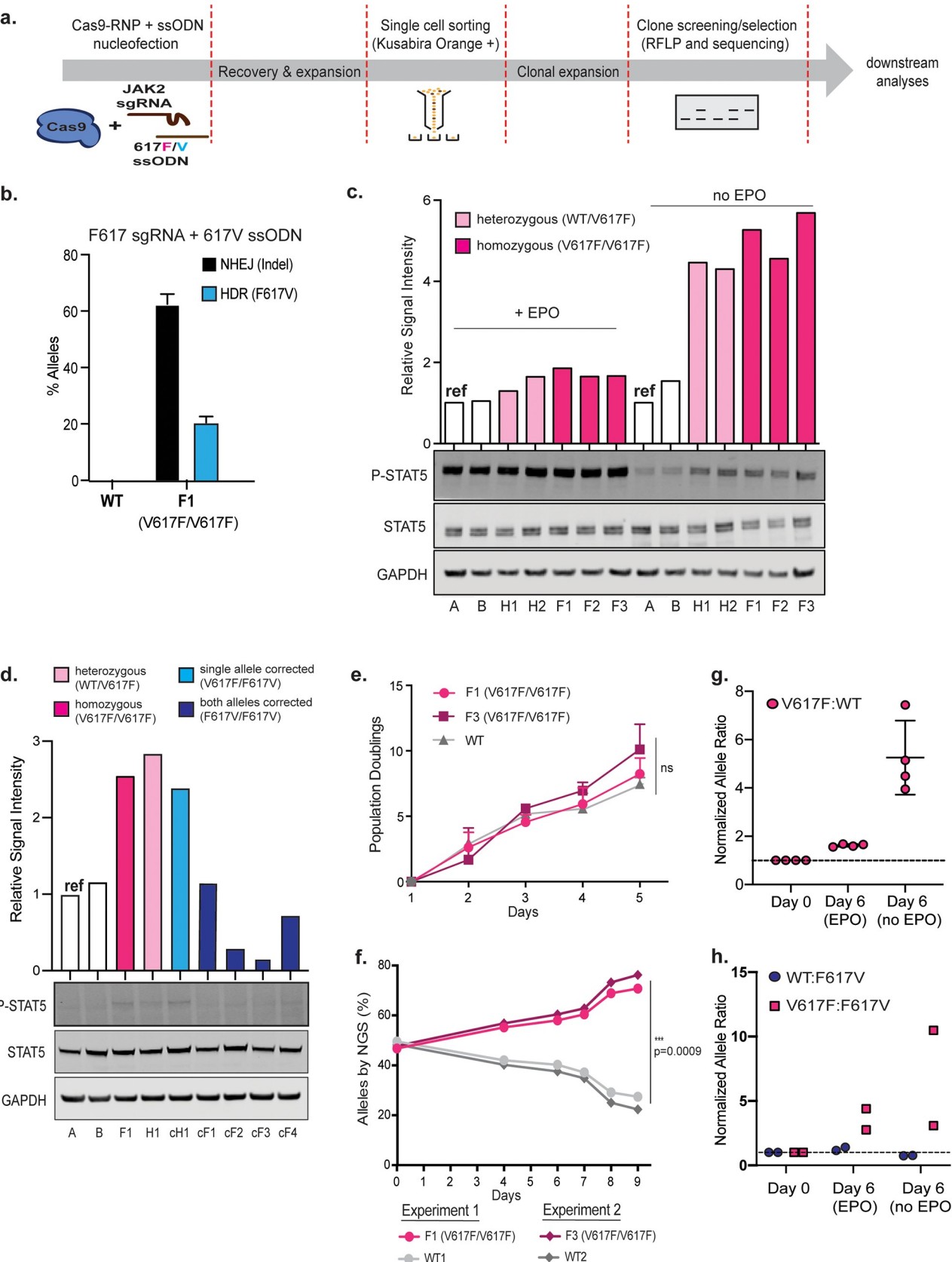

**Fig 2. Characterizing the V617F mutation in HUDEP-2s. (a)** Experimental layout for generating clonal allelic series of JAK2 V617F and F617V alleles. Nucleofected HUDEP-2 cells are recovered for 3 days and then sorted at single cell densities into a 96 well plate based on the expression of Kusabira Orange, a marker indicative of HUDEP-2 cell viability. The isolated single cells are expanded for two weeks and subsequently screened for either V617F or F617V alleles by BsaXI restriction digest. Selected clones are validated by TA-cloning and Sanger sequencing. **(b)** HUDEP-2 WT and V617F cells were nucleofected with Cas9-RNP complexed with F617-sgRNA and the 617V ssODN. NHEJ- and HDR-mediated editing outcomes were assessed by amplicon sequencing and analyzed by Interference of CRISPR Editing (ICE) software. Data from n = 4 independent biological replicates with mean±SD graphed. **(c)** Representative immunoblot and signal intensity quantification show elevated phosphorylated STAT5 (P-STAT5) expression in homozygous and heterozygous JAK2 V617F HUDEP-2 clones. Signals were normalized to STAT5 and GAPDH. White bars, JAK2 WT clones; light pink bars, JAK2 V617F heterozygous clones; magenta bars, JAK2 V617F homozygous clones. **(d)** Representative immunoblot and signal intensity quantification show reduced P-STAT5 levels following reversion of the JAK2 V617F mutation back to WT. Signal intensity calculations were normalized to STAT5 and GAPDH. White bars, JAK2 WT clones; light pink bar, JAK2 V617F heterozygous clone; magenta bar, JAK2 V617F homozygous clones; light blue bar, HUDEP clone with single allele corrected (genotype V617F/F617V); dark blue bars, HUDEP clones with both alleles corrected (genotype F617V/F617V) **(e)** Growth curve depicting cumulative population doublings of HUDEP-2 clones measured for 5 days. F1 and F3 are V617F homozygote clones. Data from n = 5 independent biological replicates. Mean of all experiments ± SD shown. **(f)** Competitive/co-culture growth assay using HUDEP-2 WT and V617F clones, showing significant outgrowth of V617F clones. Equal proportions of WT and V617F clones were co-cultured in HUDEP-2 expansion media and allelic frequencies were analyzed at 6 time-point spanning 9 days by amplicon-NGS. Circles denote co-culture of F1 and WT1. Diamonds denote co-culture of F3 and WT2. Data is from n = 2 biologically independent replicates. Error bars indicate SD.*: p<0.05 by paired t-test. **(g)** Co-culture of WT and homozygous V617F clones grown in the presence or absence of EPO for 6 days. Amplicon-NGS time points taken at Day 0 and Day 6. Ratio of V617F:WT alleles detected in culture at day 6 were normalized to day 0 and graphed. Data is from 4 biologically independent experiments each sequenced in technical triplicate. **(h)** Co-culture of either WT and homozygous F617V corrected clone (circle) or homozygous V617F and corrected F617V clones (square). NGS time points taken at Day 0, and Day 6 with and without EPO. Ratio of WT:F617V or V617F:F617V alleles detected in culture at day 6 were normalized to day 0 and graphed. Data is from 2 biologically independent experiments. Mean±SD graphed.

wild type (**Fig 2B**). We isolated 'corrected' clones using a similar pipeline described above, screening single cell clones for the reintroduction of the BsaXI site and confirming genotypes by Sanger sequencing (**S2D Fig**). We included one heterozygous corrected clone (cH1) and multiple homozygous corrected clones (cF1, cF2, cF3 and cF4) in several subsequent analyses.

JAK2 preferentially binds the EPO receptor, predominantly expressed on the surface of immature erythroid cells and it primarily signals through STAT5 and is essential for the production of red blood cells [18, 38–41]. Overexpression of JAK2 V617F leads to increased phosphorylation of STAT5 [9, 21]. We characterized levels of phosphorylated STAT5 in the presence of EPO and observed a mild increase in both heterozygote and homozygote V617F clones relative to WT (**Fig 2C**).

To determine if the V617F allele promotes cytokine-independent signaling, we cultured WT and edited clones without EPO for 1 day. In all genotypes, basal STAT5 phosphorylation was lower in the absence of EPO. But clones with the JAK2 V617F allele had greatly elevated levels of phosphorylated STAT5 relative to wild type clones (**Fig 2C and S2E Fig**). Homozygous JAK2 V617F clones appeared to have only slightly higher levels of phosphorylated STAT5 as compared to the heterozygous clones. Analysis of JAK2 activation in the corrected F617V clones showed that the heterozygous clone with one remaining V617F allele still displayed STAT5 phosphorylation when cultured in the absence of EPO, while homozygous corrected clones appeared similar to WT cells showing only basal levels of STAT5 phosphorylation (**Fig 2D**). These data suggest that the V617F allele is haplosufficient at the endogenous locus in hyperactivating JAK2 signaling and that our mutation-specific reagents are able to revert hyperactive JAK2 signaling back to WT levels.

We also observed an increase in STAT1 phosphorylation in homozygous and heterozygous JAK2 V617F clones as compared to WT clones, both with and without EPO (**S2F Fig**). However, in contrast to STAT5, levels of STAT1 phosphorylation were not increased above WT in the presence of EPO, consistent with STAT5's role as the main signal transducer of JAK2 activation in hematopoietic cells.

JAK2 V617F is associated with hyperproduction of differentiated cells in MPN patients. To determine whether the JAK2 V617F mutation confers a growth advantage on its own, we tracked population doublings and viability of two homozygous JAK2 V617F clones (F1 and

F3) and two WT clones maintained in separate culture, all in the presence of EPO. The JAK2 V617F homozygous clones exhibited a very modest increase in growth rate over WT clones in five independent experiments and were not statistically significant (**Fig 2E**). However, the JAK2 V617F mutation is somatically acquired rather than germline encoded. To recapitulate a mixed milieu of WT and mutant cells, we designed a co-culture experiment where WT and homozygous JAK2 V617F clones were grown together and allele prevalence was tracked over time by amplicon-NGS. Immediately after mixing we confirmed equal proportions of WT and V617F alleles (**Fig 2F**). Within nine days the allele distribution became dramatically skewed such that V617F constituted 80% of the alleles (**Fig 2F**). These data suggest that acquisition of the JAK2 V617F mutation leads to a competitive growth advantage over WT cells in a mixed *ex vivo* setting.

Since JAK2 V617F led to increased STAT5 phosphorylation in the absence of EPO, we asked whether the V617F mutation also conferred a cytokine-independent growth advantage. However, unlike CD34+ HSPCs, erythroid progenitors require EPO during proliferation and differentiation, and we found EPO to be essential for HUDEP-2 cells during longer-term culture. JAK2 V617F clones persisted slightly longer in EPO-free culture than WT, but this difference was not statistically significant (**S2C Fig**).

Since we previously saw marked differences between isolated and co-culture phenotypes in the presence of EPO (**Fig 2F**), we asked whether co-culture might reveal growth differences in the absence of EPO. We performed this mixed culture experiment in a relatively short time frame, since we found that prolonged culture (> 6 days) without EPO was not possible regardless of genotype. A 1:1 mixed population of a WT clone and JAK2 V617F clone were cultured in expansion media with or without EPO. HUDEP-2s carry an expression cassette for the fluorescent protein Kusabira-Orange, which we used to exclude dead cells from analysis after six days in culture (**S2G Fig**). Since Kusabira-Orange with an absorption maximum at 548nm and an emission peak of 561nm can be detected in the PE channel without any autofluorescence detected from dead/dying cells in the same channel, it served as a good marker to select for healthy and live cells. Allele frequencies were measured by amplicon-NGS as a proxy for the abundance of cells harboring each allele. In the co-culture setting, we found that JAK2 V617F was over-represented in the presence of EPO, as before. The abundance of the JAK2 V617F allele was markedly enhanced in the absence of EPO (**Fig 2G**). We normalized the ratio of V617F:WT alleles at day 6 to the starting ratio at day 0 and found that JAK2 V617F was over-represented $1.6 \pm 0.1$ fold in the presence of EPO, and this was further enhanced to $5.3 \pm 1.5$ fold in the absence of EPO.

We performed further co-culture assays to determine if correcting the V617F mutation reverses its competitive growth advantage. A 1:1 mixed population of WT clone and JAK2 F617V clone, or JAK2 V617F clone and JAK2 F617V clone, were cultured in expansion media with or without EPO. When cultured with EPO for six days, the F617V corrected alleles were 2.8 fold less prevalent than V617F alleles (**Fig 2H**). Reversion of the V617F proliferative advantage became even more apparent when cells were cultured without EPO and the V617F alleles were 10.5 fold more prevalent than the corrected F617V alleles (**Fig 2H**). When WT and F617V cells were cultured together, allele prevalence remained the same after 6 days in culture with EPO (**Fig 2H**).

MPN leads to excess production of erythroid lineages, and we investigated whether the V617F mutation altered the ability of HUDEP-2s to differentiate into erythroblasts. We differentiated WT, heterozygous and homozygous JAK2 V617F clones in erythroid induction medium containing EPO and assessed the efficiency of differentiation after 5 days by flow cytometry for the cell surface marker glycophorin A (GlyA). Surprisingly, we found no difference in differentiation between WT and either heterozygous or homozygous JAK2-V617F

HUDEP-2 clones. All clones completed differentiation by day 5, regardless of genotype (**S2H Fig**). These data suggest that in conditions with ample cytokine signaling, the JAK2 V617F allele does not confer increased *in vitro* erythroblastic differentiation relative to WT cells.

## JAK2 V617F promotes EPO-independent growth in human peripheral mobilized CD34+ HSPCs

Human CD34+ HSPCs are clinically relevant for MPN physiology, and unlike HUDEP-2 cells, do not require EPO to proliferate in culture. Furthermore, the ability of HSPCs to differentiate into divergent lineages provides an opportunity to interrogate how the V617F mutation alters *in vitro* differentiation beyond the erythroid lineage.

We first tested whether bulk editing of CD34+ cells leads to efficient editing in all stem/progenitor subpopulations. We nucleofected HSPCs with sgRNA4-RNP and the 617F ssODN, cultured the cells for 3 days, immunophenotypically sorted several cell subpopulations by fluorescent activated cell sorting (FACS) and measured editing outcomes by amplicon-NGS. HSCs, multipotent progenitors (MPPs), common myeloid progenitors (CMPs) and multipotent lymphoid progenitors (MLPs) were isolated by characteristic surface markers (**S3A Fig**). The composition of most subpopulations in culture did not markedly change from the time of edit to the time of harvest 3 days later (**S3B Fig**). Intriguingly, we found that indel frequencies were similar for all subpopulations but HDR frequencies varied greatly (**S3C Fig**). The disparity of HDR in different progenitors suggest certain CD34+ HSPC subpopulations may be more amenable to delivery of editing reagents, or alternatively are differentially able to perform HDR. The latter hypothesis is supported by a recent publication and a preprint investigating HDR in long-term HSCs versus more differentiated progenitors [42–44].

To eliminate biased editing outcomes arising from differential editing within HSPC subpopulations, we modified a recently described protocol to isolate and edit only immunophenotypic HSCs (**Fig 3A**) [43]. We cultured CD34+ HSPCs in cocktail of CC110 and additives as previously described to stimulate HSC expansion (see Methods for details) [45, 46]. Following expansion, we immunophenotypically isolated HSCs by FACS and immediately nucleofected them with a Cas9-sgRNA4 RNP and an ssODN to generate either the JAK2 V617F or V617V allele. As an additional control for the effects of editing itself, we targeted the beta-globin gene (*HBB*), which is unrelated to MPN. For *HBB* targeting, we used a previously validated and highly effective sgRNA and ssODN combination (approximately 60% NHEJ and 20% HDR) designed to introduce the causative mutation involved in sickle cell disease [47]. These *HBB* editing reagents served as a benchmark for the performance of our altered cultured conditions and additional FACS sort. Following electroporation, cells were allowed to recover for two days, after which a 'input' sample was taken to determine editing efficiencies, while the remainder of cells were subjected to additional assays. Using the modified editing workflow, rates of HSC editing at *HBB* by amplicon-NGS were comparable to previous studies (**Fig 3B**). *JAK2* editing was similar whether using V617F (25.9 ± 0.8%) or V617V (24.5 ± 0.03%) reagents (**Fig 3B**).

As MPN patients exhibit hyperproliferation of myeloid and erythroid cells, we asked whether the JAK2 V617F mutation is sufficient to skew early *in vitro* HSC differentiation. We maintained bulk edited HSCs in SFEMII stem cell expansion media supplemented with CC110 cytokine cocktail and allowed the cells to spontaneously differentiate in culture for four days after editing. We then used flow cytometry to separate HSCs, multipotent progenitors (MPPs), multipotent lymphoid progenitors (MLPs), common myeloid progenitors (CMPs), megakaryocyte-erythroid progenitors (MEPs), and B/NK cells as described in the methods (**S3A Fig**). We found that the subpopulation distribution of progenitors was similar for the V617F and V617V edited pools (**S3D Fig** and **S1D Table**).

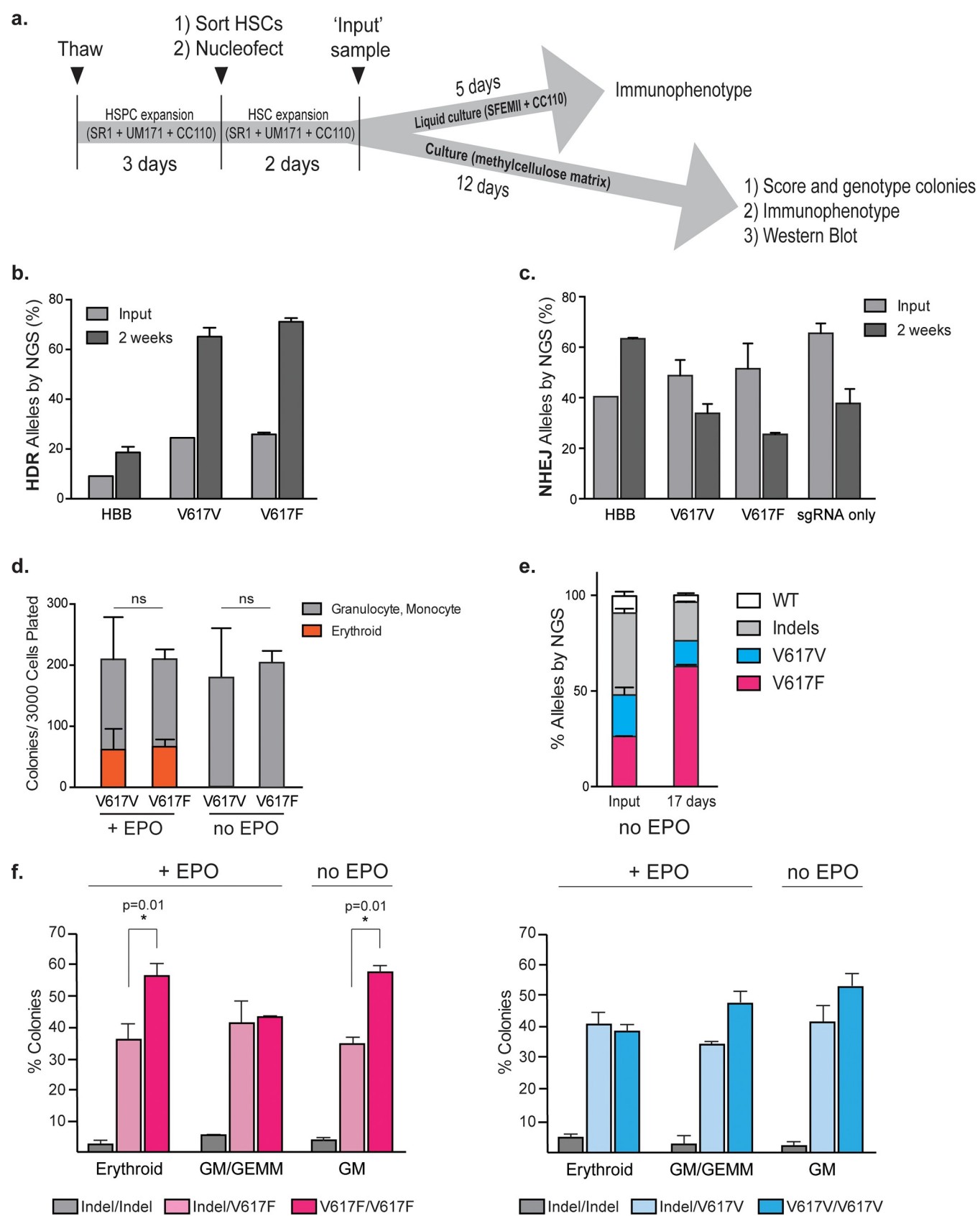

**Fig 3. Identifying the enrichment of the V617F-edited alleles in CD34+ hematopoietic stem cells. (a)** Experimental layout for editing LT-HSCs. HSPCs were thawed and expanded for 3 days in SR1, UM171 and CC110. HSCs were sorted based on CD34+ CD38- CD45RA- CD90+ and nucleofected with appropriate editing reagents. Two days following electroporation an 'input' sample was taken to determine editing percentages. Cells were either maintained in liquid culture for immunophenotyping or plated in methylcellulose for further analyses. **b.** HSCs were nucleofected with Cas9-RNP and ssODNs encoding HBB, V617F or V617V mutations and were plated on methylcellulose as depicted in **(a)**. HDR-mediated outcomes were assessed by NGS, two days (input) or two weeks after electroporation. Data from n = 3 independent biological replicates. Mean±SD shown. **(c)** NHEJ-mediated outcomes of cells in **(b)** were assessed by NGS, two days (input) or two weeks after electroporation. Data from n = 3 biological replicates. Mean±SD shown. **(d)** Pools of V167F- or V617V-edited HSCs were plated on methylcellulose with or without EPO and were scored as granulocyte, monocyte or erythroid based on morphology 14 days after plating. Data from n = 4 (for V617F) or n = 3 (for V617V) independent biological replicates. Mean±SD shown. **(e)** V617F- and V617V-edited HSCs were mixed and plated on methylcellulose. Input sample was taken directly after mixing prior to plating on methylcellulose. All resulting colonies were harvested two weeks later and processed for amplicon-NGS. Data from n = 4 biological replicates. Mean±SD shown. $^*$:p<0.05 by unpaired t-test. **(f)** Single colonies were phenotyped as erythroid, GEMM, or GM and then genotyped by amplicon-NGS. V617F homozygous colonies were preferentially selected in the absence of EPO. Data from n≥3 biological replicates. Mean±SD shown. $^*$:p<0.05 by unpaired t-test.

We therefore asked whether JAK2 V617F affects more terminally differentiated populations as opposed to progenitors. Because terminal differentiation is difficult to accomplish in liquid culture, we plated edited HSCs in methylcellulose and allowed them to form colonies over two weeks. We then either scraped and collected all colonies for Western blotting and flow cytometry, or morphologically classified single colonies and genotyped each individual colony by amplicon-NGS.

Edited HSCs had input HDR frequencies of approximately 24.5 ± 0.03% JAK2 V617V and 25.9 ± 0.8% V617F (**Fig 3B**). After edited HSCs were grown on methylcellulose for two weeks, we found an increase in HDR to approximately 65.1 ± 3.6% for V617V alleles and 71.1 ± 1.5% for V617F alleles (**Fig 3B**). We observed a corresponding decrease in indel frequencies from 48.8 ± 6.2% to 33.8 ± 3.7% for V617V, and from 51.5 ± 10% to 25.5 ± 0.6% for V617F (**Fig 3C**). Notably, we did not observe a similar increase in HDR and decrease in indels for editing at the *HBB* locus, suggesting that the act of editing itself does not affect HDR vs indel allele prevalence over time. Based on our earlier finding that a majority of the indels from sgRNA-4 (located in the coding region) result in frameshifts that introduce premature stop codons or otherwise disrupt protein function (**Fig 1E**), we hypothesized that *JAK2* indels could affect the fitness of the differentiated colonies and be negatively selected. We nucleofected HSCs with sgRNA-4 alone but no ssODN and found a marked decline in indel alleles over time, from 65.5 ± 3.9% at the time of edit to 37.8 ± 5.7% after two weeks (**Fig 3C and S3E Fig**). Intact coding alleles (V617V or V617F) appeared haplosufficient since genotyping individual colonies revealed approximately 35% indel/HDR genotypes after two weeks, but less than 3% indel/indel genotypes (**Fig 3F**). We subsequently used the V617V HDR allele as a negative control for the effect of the V617F HDR mutation.

Acquisition of the JAK2 V617F mutation in HSCs or progenitors is proposed to be the transforming event that leads to eventual clonal outgrowth of the parental cell population [48, 49]. To determine whether acquisition of the JAK2 V617F mutation causes a greater number of colonies to form in a methylcellulose colony-forming unit (CFU) assay, we quantified the number of colonies arising from JAK2 V617F- and V617V- edited HSCs in the presence and absence of EPO stimulation. Since HSPCs have been implicated as harboring the V617F mutation in MPNs, we asked how the acquisition of the mutation at the endogenous locus affects differentiation of the cells to erythroid cells, which are most relevant for PV. However, we did not observe any difference in the total number of the colonies formed (**Fig 3D**). JAK2 V617F was positively selected during co-culture with WT cells in a HUDEP-2 background cultured in the absence of EPO, and we performed a similar co-culture experiment using HSCs. We combined equal numbers of JAK2 V617F- and V617V-edited HSCs, took a sample for the 'input' time point, and plated the edited HSCs on methylcellulose without EPO. After 17 days, we scraped the resulting colonies and determined overall allele prevalence by amplicon-NGS.

Analysis of the 'input' time point showed the mixed pool of cells had equal representation of JAK2 V617F (26.2 ± 0.2%) and V617V (21.7 ± 3.9%) alleles (**Fig 3E**). After 17 days we found strong positive selection of the V617F allele (62.7 ± 0.1%) over the V617V allele (13.5 ± 0.01%). Thus the JAK2 V617F allele exerts a competitive growth advantage in human HSCs and progenitors relative to the wildtype allele.

Because higher JAK2 V617F allele burden is associated with the development of clinical MPNs [50], we investigated whether edited colonies that were homozygous for V617F gave rise to a greater number of erythroid and granulocyte-macrophage colonies than heterozygous colonies. We edited HSCs, grew them on methylcellulose with and without EPO and used amplicon-NGS to individually genotype a total of 432 colonies (144 Erythroid, 144 GM/GEMM, and 144 GM). Very few colonies of any cell type harbored homozygous indel/indel mutations, further indicating that homozygous disruption of JAK2 is negatively selected during HSC differentiation. We found no significant allelic advantage for JAK2 V617V in any cell type, either with or without EPO (**Fig 3F**). In the presence of EPO, we found a statistically significant increase in erythroid colony formation in JAK2 V617F homozygotes (53.6 ± 4.4%) relative to V617F heterozygotes (36.3 ± 5.2%) (**Fig 3F**). This advantage was not reflected in GM/GEMM colonies. In the absence of EPO, we observed a significant increase in the percentage of GM colonies that were JAK2 V617F homozygotes (56.5 ± 3.7) as compared to V617F heterozygotes (31.3 ± 0.7%) (**Fig 3F**).

## JAK2 V617F is sufficient to induce partial erythroid differentiation

To characterize the lineages of colonies derived from HSCs separately edited with JAK2 V617F or V617V reagents, we scraped colonies formed on methylcellulose and immunostained for surface markers specific to different lineages (CD19, CD56, CD33, CD14, CD41, CD71, GlyA). When colonies were grown in the presence of EPO, we found limited colonies from the myeloid and lymphoid lineages and no difference between JAK2 genotypes (**Fig 4A and 4B**). Most colonies developed into erythroid CD71+ and GlyA+ cells. There was no difference in the fraction of cells expressing CD71 based on JAK2 editing (V617F 70.4% CD71+ and V617V 65.6% CD71+). However, we observed a significant increase in GlyA+ mature erythrocytes derived from V617F-edited HSCs (58 ± 19.5%) as compared to V617V-edited HSCs (26.4 ± 8.8%) (**Fig 4A**).

When colonies were grown without EPO, the majority of the cells were early myeloid progenitors (CD33+) and there were also a significant number of monocytes (CD14+) and lymphoid lineages (CD19+ or CD56+). There was no difference in abundance of these cells between JAK2 genotypes (**Fig 4B**). We found almost no colonies from erythroid lineages in the V617V-edited HSCs, indicative of the requirement of EPO for erythroid differentiation. By contrast, the fraction of cells expressing GlyA markedly increased to 6.6 ± 1.4% in JAK2 V617F edited HSCs from 0.8 ± 0.6% in V617V edited HSCs (**Fig 4B**), highlighting that JAK2 V617F mutation renders the cells insensitive to EPO.

We finally examined STAT5 phosphorylation and β-globin expression during erythrocyte differentiation of edited HSCs. Unlike in clonally derived HUDEP-2 cells, HSCs edited for JAK2 V617F did not exhibit an increase in phosphorylated STAT5 (**Fig 4C**). This may be due to lower allele burden, since only ~25% of the bulk HSC population alleles are V617F but HUDEP-2 clones harbor either 50% (heterozygous clones) or 100% (homozygous clones) V617F. Colonies that were driven to erythrocyte differentiation by EPO showed high β-globin levels in all genotypes (**Fig 4C**). In the absence of EPO, we found that JAK2 V617F editing was sufficient to induce high expression of β-globin (**Fig 4C**). This was especially striking considering that FACS immunophenotyping indicated that only 6.6 ± 1.4% of cells are of erythroid

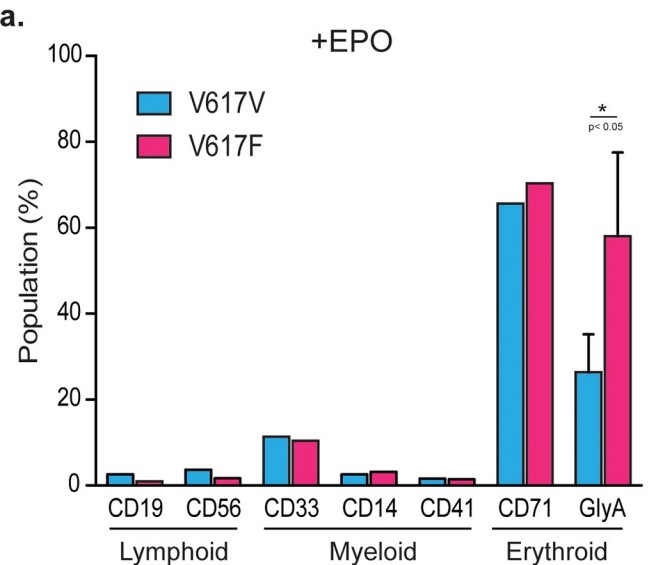

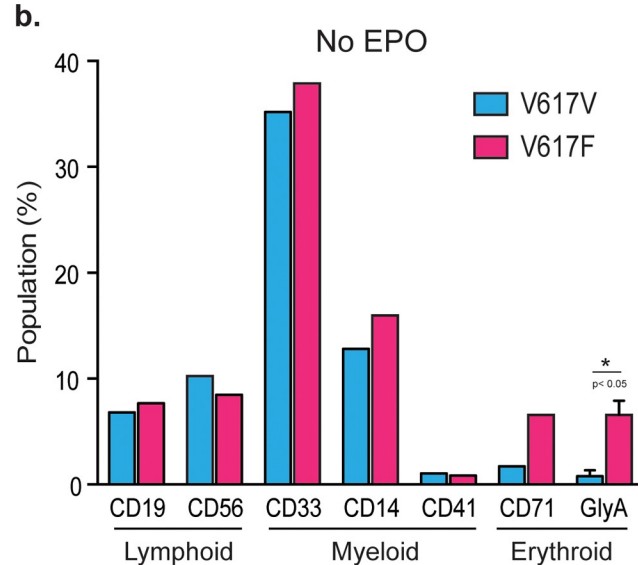

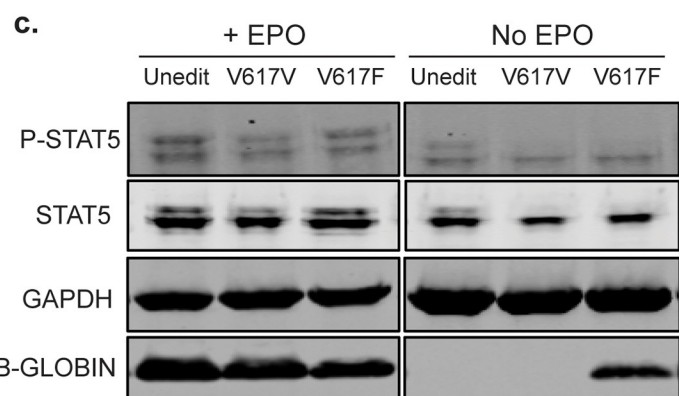

**Fig 4. Characterizing the effect of the V617F mutation in CD34+ hematopoietic stem cells. (a, b)** Immunophenotypic analysis of methylcellulose colonies cultured with or without EPO. All colonies were scraped, pooled and stained the following surface markers. Lymphoid: CD19 or CD56; Myeloid: CD33, or CD14 or CD41a; Erythroid: CD71 or GlyA. Data from n = 1 for all except GlyA samples. GlyA data from n = 3 independent biological replicates. Mean±SD shown. *:p<0.05 by paired t-test. **(c)** Immunoblot showing the elevated expression of B-globin in V617F edited cells grown in methylcellulose without EPO. No difference was observed in P-STAT5 expression across all samples. GAPDH served as a loading control.

lineages (**Fig 4B**). Taken together, our data suggest that acquisition of the JAK2 V617F mutation at the endogenous locus is sufficient to promote terminal differentiation of erythroid progenitors when EPO signaling is either abundant or limiting (**Fig 4A and 4B**).

## Discussion

The JAK2 V617F mutation is highly prevalent in MPNs and has been proposed to be an initiating event for the disease. Studies have shown that transplanting a single V617F-expressing LT-HSC into a lethally irradiated mouse can give rise to MPN. However, in this case disease was only initiated in 4/17 (24%) of recipient mice [51]. Other studies suggest that while JAK2 V617F amplifies hematopoietic progenitors by conferring augmented growth and differentiation, it remains insufficient in to engender the full blown disease [52–54]. The JAK2 V617F mutation can also be frequently detected in the peripheral blood of healthy individuals who do

not yet have symptoms of hematological disease [55, 56]. Single cell DNA and RNA sequencing of MPN patients has highlighted that acquiring additional mutations, such as those in *TET2* and *EZH2*, in a particular order play a role in the development of specific subsets of MPN [31]. The link between the prevalence of the JAK2 V617F allele and initiation of MPNs has thus recently become less clear, and it is possible that JAK2 mutation is associated with clonal hematopoiesis of indeterminant potential (CHIP) rather than causative of disease.

Using genome editing to generate JAK2 V617F and V617V mutations in HUDEP-2s and primary human HSCs, we found that the JAK2 V617F allele does not dramatically affect cultured cell growth or colony formation of mutant cells on their own. The allele therefore does not behave like a frankly transforming mutation. Instead, we found that the JAK2 V617F allele confers a proliferative advantage during *in vitro* co-culture experiments, such that mutant cells outgrow WT cells. The progressive outgrowth of V617F-positive HUDEP-2 clones and HSC colonies lends support to the idea that JAK2 is sufficient to engender clonal expansion.

Previous studies have shown that the STAT5 transcription factor plays a critical role in MPN pathogenesis [57–59]. While we have also observed that JAK2 V617F confers STAT5 activation, we found that a single V617F allele was sufficient to increase STAT5 signaling in the absence of EPO, highlighting the cytokine-independent signaling mediated by JAK2-V617F. Correspondingly, the expansion of the JAK2 V617F allele was markedly enhanced in EPO-free conditions in comparison to EPO-containing cultures. We furthermore found that acquisition of the V617F mutation promotes colony formation and the appearance of markers of terminal erythrocyte differentiation, even in the absence of cytokine signaling. Recent transcriptomic analysis of MPN patient cells harboring CALR mutations similarly found that committed myeloid progenitors exhibit increased proliferation signatures relative to more primitive progenitors [60]. The increased fitness of MPN-associated alleles may therefore be most manifest in conditions of limited cytokine signaling, where terminal differentiation is normally reduced. While EPO levels increase during healthy aging, EPO tends to be low in most PV patients. However, the precise cytokine environment associated with MPN onset can be strikingly varied and requires further investigation [61–64]. It will also be critical to further examine expansion and differentiation of MPN allele-harboring human HSPCs using xenograft assays in mouse models that allow for enhanced erythropoietic and megakaryocytic lineage outputs, such as humanized NSGW41 [65]. Such experiments will also be important to further test the competitive advantage of V617F CD34+ HSPCs in a more complex milieu.

We targeted a single MPN-associated mutation, but future work could attempt to simultaneously edit multiple genes implicated in MPN. This would more accurately model the co-occurrence of multiple mutations in individuals with MPN. However, such an approach may be complicated by multiple factors: the heterogeneous composition of edited alleles in pools of CD34+ cells, the possibility of translocations between co-targeted sites, and the lack of molecular selection for co-edited cells. However, recent work performing simultaneous single-cell sequencing of DNA and RNA from MPN patients suggests a potential avenue for post-hoc separation and phenotyping of multi-gene edited cells versus cells with unintended genetic outcomes [31, 60].

Because MPNs originate from self-renewing stem or progenitor populations, the only existing curative therapy is to replace mutated cells with wild type HSCs via allogeneic transplant. We developed an F617-sgRNA and V617V ssODN that specifically target and correct disease alleles while sparing wild type alleles. While our primary intention with the F617V reversion reagents was for it to act as a "clean" control, we also wanted to knowledge the theoretical utility of these reagents together with the concerns that may present a practical barrier. For example, by editing wild type cells with F617V reagents serving as a "clean" control, we show that the phenotypes observed are not a result of editing itself (e.g., due to p53 activation). A genome

editing based autologous transplant could be a potential treatment for MPN, thereby reducing the risks associated with allogeneic transplants especially for elderly patients. Therefore, we reverted the V617F mutation (itself made using genome editing), which is akin to an endogenous rescue experiment. Much work remains to be done regarding the safety of therapeutic gene editing, but there is accumulating evidence that causative mutations in monogenic hematopoietic disorders can be repaired in patient HSCs [1, 47, 66–69]. Curing overtly oncogenic disorders using genome editing is difficult due to the need for complete allele conversion, since cells that escape the edit may eventually recover to take over the population. However, since the increased co-culture fitness of the JAK2 V617F mutation is a gradual event, it may be possible that even incomplete editing could lead to extended disease remission.

## Methods

### K562 cell culture

K562 cells, acquired from the UC Berkeley Cell Culture Facility, were maintained in IMDM supplemented with 10% fetal bovine serum (FBS), 1% sodium pyruvate, 1% non-essential amino acids and 100ug/ml penicillin-streptomycin. Cells were tested for mycoplasma contamination using enzymatic (Lonza, Basel, Switzerland) and PCR-based assays (Bulldog Bio).

### HUDEP-2 cell culture

HUDEP-2 cells were acquired from the Nakamura Lab at the RIKEN BioResource Center in Japan and were cultured in StemSpan SFEM (Stem Cell Technologies) supplemented with 50ng/ml SCF (R&D Systems), 3IU/ml EPO (KIRIN or Peprotech), $10^{-6}$M Dexamethasone (Dex, SIGMA-D2915), and 1ug/ml Doxycycline (Dox) (TAKARA Bio), and 100ug/ml penicillin-streptomycin, unless otherwise noted. Note: HUDEP-2 cells can proliferate in the absence of Dex for a short period, however we cultured the cells in Dex-containing medium for a long-term culture. SCF, EPO, and Dox are essential for culturing HUDEP-2 cells. Cells were split 1:6 or 1:10 every 3 or 4 days and density was maintained below $10^6$ cells/ml as high cell density deteriorated cell viability. Cells were tested for mycoplasma contamination using enzymatic (Lonza) and PCR-based assays (Bulldog Bio).

### HUDEP-2 cell differentiation

HUDEP-2 cell differentiation was induced using a modified version of protocol adapted from [37]. For induction of differentiation of cells to more mature erythroid cells, HUDEP-2 cells were cultured in IMDM (Thermo Fisher) supplemented with 2% FBS, 3% Human Serum Albumin (Sigma-H4522), 3IU/ml EPO (KIRIN, or Peprotech), 10ug/ml Insulin (Sigma-I2643), 500 – 1000ug/ml Holo-Transferrin (Sigma-T0665), 3U/ml Heparin (Sigma-H3149). It takes 5 days for HUDEP-2s to fully differentiate.

### CD34+ primary cell culture

Frozen human mobilized peripheral blood CD34+ HSPCs were purchased from AllCells and thawed according to manufacturer's instructions. CD34+ cells were cultured in StemSpan SFEMII (Stem Cell Technologies) supplemented with StemSpan CC110 cocktail (Stem Cell Technologies) containing SCF, TPO and Flt3L, as well as the additives 35nM UM171 (Stem Cell Technologies) and 750nM StemRegenin1 (SR1; Alichem) which have been previously described to stimulate HSC expansion [46, 70], unless otherwise noted.

### *In vitro* transcription of gRNA

gRNA was synthesized by assembly PCR and *in vitro* transcription as previously described [47]. Briefly, primers with the T7 promoter, desired protospacer and a truncated constant region of the tracrRNA were annealed and amplified with Phusion high-fidelity DNA polymerase (New England Biolabs) to produce the chimeric sgRNA template for transcription. *In vitro* transcription was carried out by the HiScribe T7 High Yield RNA Synthesis kit (New England Biolabs). Resulting transcriptions products were treated with DNAse I, and RNA was purified by treatment with a 5X volume of homemade solid phase reversible immobilization (SPRI) beads and elution in RNAse-free water. sgRNA concentrations were determined by fluorescence using the Qubit dsDNA HS assay kit (Life Technologies).

### RNP assembly and electroporation of K562s, HUDEP-2s and HSPCs

Cas9 RNP assembly and cell electroporation were carried out as previously described [47]. Briefly, 75pmol of Cas9 was gradually mixed into Cas9 buffer (20mM HEPES; pH 7.5, 150mM KCl, 1mM $MgCl_2$, 10% glycerol and 1mM TCEP) containing 75pmol of either IVT gRNA for K562s or synthetic sgRNA (Synthego) for HUDEP-2s and HSPCs, constituting a total volume of 7.5ul. The RNP mixture was incubated at room temperature for 15 minutes to allow RNP complex to form. $1x10^5$ cells were washed once with PBS and then resuspended in 21.5μl of SF or P3 electroporation buffer (Lonza), for K562s and HUDEP-2s/HSPCs, respectively. 1μl of 100μM ssODN template was added to 7.5μl of RNP mixture, and the combined mixture was added into a Lonza 4d strip nucleocuvette. Immediately, 21.5μl of cell suspension was added into a Lonza 4d strip well containing the RNP and ssODN mix, and the total mixture was mixed by gentle pipetting. The cells were electroporated with program FF-120 for K562s, DD-100 for HUDEP-2s and ER-100 for HSPCs, and subsequently transferred to a culture dish containing pre-warmed media. Editing outcomes were measured 3 or 17 days post-electroporation by Next Generation Amplicon Sequencing (see later in the article).

### Genomic DNA extraction

10K-400K edited cells were collected, washed once in PBS and resuspended in 20–50μl Quick-Extract DNA Extraction Solution (Lucigen). The reaction mixture was incubated at 65°C for 10 minutes and then 95°C for 2 minutes. The extract was spun at maximum speed in a microcentrifuge for 2 minutes, and the supernatant was used for PCR amplification.

### T7E1 assay and restriction digest

Cells were harvested 48–72 hours post-electroporation and genomic DNA was extracted. Approximately 500bp of the edited locus was PCR amplified (primers in **S1B Table**). The PCR product was denatured at 95°C for 5 minutes and re-annealed slowly decreasing the temperature to 25°C in a thermocycler at a ramp rate of -2°C/sec from 95°C to 85°C, and -0.1°C/sec from 85°C to 25°C, and subjected to T7 endonuclease I (New England Biolabs) digestion for 30 minutes at 37°C. The digestion product was run out on an agarose gel. Efficiency of sgRNA/Cas9-mediated cutting was estimated by the amount of digestion by the T7E1 endonuclease.

### Generation and screening of mutant HUDEP-2 clones

72 hours post-electroporation, edited pools of HUDEP-2 cells were harvested and subjected to fluorescence-activated cell sorting (FACS). Cells expressing moderate to high levels of Kusabira Orange, a marker gene indicative of viable HUDEP-2s, were sorted into a 96 well plate,

containing pre-warmed media, at single cell densities, grown to confluency and split into duplicate plates. For the screening process, one set of the duplicated plates was used for genomic DNA extraction using QuickExtract DNA Extraction Solution (Lucigen). 1–2μl of extracted genomic DNA in a total PCR reaction volume of 25μl were amplified in a 35 cycle PCR using a locus-specific primer set (primer set 1) and AmpliTaq Gold 360 Master Mix (Applied Biosystems) (**S1B Table**). The PCR amplicons were subjected to restriction digestion using BsaXI (New England Biolabs) for 60 minutes at 37˚C, and the digested products were run on a 1.5% agarose gel to visualize either the appearance (for WT and F617V clones) or disappearance (for V617F clones) of the digested products. Clones selected from the restriction digest screen were further verified by Sanger sequencing. The edited locus was PCR-amplified and TA-cloned using the TOPO-TA cloning kit (Invitrogen), and a minimum of 10 white colonies derived through ampicillin-Xgal selection were submitted for sequencing.

## HUDEP-2 growth and viability assay

30,000 cells were plated in HUDEP-2 expansion media containing either 3IU/ml EPO, or no EPO, unless stated otherwise. Cell numbers were counted and recorded for 5 consecutive days, using Trypan Blue exclusion for detection of viability. Live or dead cells were counted and cell viability was calculated as the following: cell viability = viable cell#/total cell #. Number of population doublings was calculated as follows: # population doublings = LN(total cell #/initial cell #)/LN(2).

## HUDEP-2 competitive/co-culture growth assay

30,000 cells of a V617F clone was combined together with 30,000 wild-type cells. The combined population was cultured in HUDEP-2 expansion media containing either 3IU/ml EPO or no EPO, unless stated otherwise. For cells cultured with EPO, aliquots of live cells were collected at days 0, 4, 6, 7, 8, and 9. For cells cultured without EPO, live cells (expressing Kusabira Orange, a marker gene indicative of viable HUDEP-2s) were collected using FACS at days 0, and 5 or 6. Genomic DNA was extracted from all collected cells and processed for downstream NGS analysis.

## Next-generation sequencing (NGS) amplicon and library preparation

50-100ng of genomic DNA from edited cells in a total PCR reaction volume of 50ul were amplified in a 30 cycle PCR using a locus-specific primer set (primer set 1) and AmpliTaq Gold 360 Master Mix (Applied Biosystems) (**S1B Table**). The PCR amplicon was purified using solid phase reversible immobilization (SPRI) beads, run on a 1.5% agarose gel to verify size and purity, and quantified by Qubit Fluorometric Quantitation (Thermo Fisher Scientific). Next, 20-50ng of the first PCR amplicon in a total PCR reaction volume of 25ul was amplified in a 12 cycle PCR using primer set 2 and PrimeSTAR GXL DNA Polymerase (Takara) (**S1B Table**). The second PCR product was SPRI cleaned and run on a 1.5% agarose gel to verify size and purity. 20-50ng of the second PCR product was subjected to the amplify-on reaction with 0.5μM forward/reverse primer pairs (primers designed and purchased through Vincent J. Coates Genomics Sequencing Laboratory (GSL) at University of California, Berkeley) (**S1B Table**) in a total PCR reaction volume of 25ul with PrimerSTAR GXL DNA Polymerase, for 9 PCR cycles. The adaptor-conjugated PCR amplicon was quantified by Qubit Fluorometric Quantitation, and a library consisting numerous edited pools of cells were multiplexed and combined at equimolar amounts. Library size and purity was verified by Bioanalyzer trace before being submitted to the GSL for paired-end 300 cycle processing using a version 3 Illumina MiSeq sequencing kit (Illumina).

## Next-generation amplicon sequencing analysis

Amplicon samples were deep sequenced on an Illumina MiSeq, using a 300bp paired-end cycle read for a depth of at least 10,000 reads. The analysis of the samples was conducted using Cortado (https://github.com/staciawyman/cortado) to HDR and NHEJ. Briefly, reads were adapter trimmed, joined into single reads, and then aligned to a reference sequence. Reads were quantified as HDR if they contained changes in the donor sequence, and as NHEJ if an insertion or deletion overlapped a 6 bp window around the cut site. Subsequently, proportions of HDR- and NHEJ-mediated repair outcomes were quantified as a percentage of total aligned reads.

## Fluorescence-activated cell sorting (FACS) or analysis

For isolation of HSCs, $1x10^6$ CD34+ HSPCs were stained with APC-Cy7-anti-CD34 (1:100) (BioLegend), PE-Cy7-anti-CD38 (1:50), PE-anti-CD90 (1:30), FITC-anti-CD45RA (1:25). All antibodies were purchased from BD Biosciences, unless otherwise noted. Samples were sorted on Aria Fusion Cell Sorter.

HSCs were characterized by CD34+ CD38- CD45RA- CD90+, multipotent progenitors (MPPs) by CD34+ CD38- CD45RA- CD90-, multipotent lymphoid progenitors (MLPs) by CD34+ CD38- CD45RA+ CD90-/lo, common myeloid progenitors (CMPs) by CD34+ CD38+ CD45RA- CD10- CD135+, megakaryocyte-erythroid progenitors (MEPs) by CD34+ CD38+ CD45RA- CD10- CD135-, and B/NK cells by CD34+ CD38+ CD45RA+ CD10-.

For isolation or immunophenotypic analysis of MPP, CMP/MEP, B/NK/G, CMP, MEP populations, $1x10^6$ CD34+ HSPCs or HSCs were stained with APC-Cy7-anti-CD34 (1:100) (BioLegend), PE-Cy7-anti-CD38 (1:50), PE-anti-CD90 (1:30), FITC-anti-CD45RA (1:25), APC-anti-CD10 (1:100) (BioLegend), PerCP-Cy5.5-anti-CD135 (1:100) (BioLegend). All antibodies were purchased from BD Biosciences, unless otherwise noted. Samples were sorted on Aria Fusion Cell Sorter or analyzed on either Aria Fusion Cell Sorter or LSR Fortessa cytometer.

For isolation or immunophenotypic analysis of cells/colonies derived from the methylcellulose assay, $1x10^6$ cells were stained with one of the following antibodies: APC-anti-CD19 (1:100), PerCP-Cy5.5-anti-CD56 (1:100), FITC-anti-CD33 (1:100), BV421-anti-CD14 (1:100), FITC-anti-CD41a (1:100) (BD Biosciences), PE-anti-CD71 (1:100), BV421-anti-CD235a (1:100) (BD Biosciences). All antibodies were purchased from BioLegend, unless otherwise noted. Samples were sorted on Aria Fusion Cell Sorter or analyzed on LSR Fortessa cytometer.

All cells were washed at least once (3 times for cells derived from methylcellulose matrix) before staining. The cells were incubated at 4˚C for 30 to 60 minutes with the indicated antibodies in staining media (PBS with 2.5% FBS added), washed twice with staining media, spun at 1200$rpm$ for 3 minutes, and supernatant was removed. The cell pellet was gently resuspended in 50–200μl of staining media for downstream analysis.

## Colony-forming unit (CFU) assay

HSCs were plated at 500 cells per 6cm dish filled with 1ml of either Methocult-Enriched (H4435, Stem Cell Technologies) supplemented with recombinant cytokines including EPO, or Methocult-Optimum (H4035, Stem Cell Technologies) without EPO, according to the manual's protocol (Stem Cell Technologies). Syringes and large bore blunt-end needles were used to dispense viscous methylcelluose medium. Methocult H4435 is formulated to support optimal growth of erythroid progenitor cells (CFU/BFU-E), granulocyte-macrophage progenitor cells (CFU-GM, CFU-G, CFU-M) and multipotential granulocyte, erythroid, macrophage, megakaryocyte cells (CFU-GEMM). Methocult H4035 without EPO is formulated to support

optimal growth of granulocyte-macrophage progenitor cells (CFU-GM, CFU-G, CFU-M). Cells were cultured at 37˚C, 5% $CO_2$, and 5% $O_2$ for 14 days before processed for downstream analysis. Note, to prevent the methocult from dehydration, additional 6cm dishes were filled with 1ml of PBS and incubated together on a 15cm dish.

## Scoring and genotyping methylcellulose colonies

14 days after plating HSCs, colonies grown with EPO were counted and scored as BFU/CFU-E or CFU-GM/GEMM. For colonies grown without EPO, colonies were scored as CFU-GM, according to the manual for Human Colony-forming Unit (CFU) Assays Using MethoCult (StemCell Technologies).

For genotyping, a single colony was picked and transferred to a 96-well PCR plate (Thermo Fisher Scientific), containing 30μl QuickExtract DNA Extraction Solution (Lucigen). A full plate containing the reaction mixture (single colony + extraction solution) was tightly sealed with a microseal, and incubated at 65˚C for 10 minutes and then 95˚C for 2 minutes. Extraction was spun at maximum speed in microcentrifuge for 2 minutes, and the supernatant was used for PCR amplification. The amplified product was further processed through the NGS library preparation and analysis pipeline as previously described.

## Immunoblotting

Cells were harvested and washed with PBS (minimum of 5 times for methylcellulose colonies). Whole-cell extract was prepared from 1X *radioimmunoprecipitation a*ssay lysis buffer (RIPA; Millipore). Extract was clarified by centrifugation at 15,000*g* for 15 min at 4˚C, and protein concentration was determined by Pierce BCA (bicinchoninic acid) assay (Thermo Fisher Scientific). 8–30μg of whole-cell extract was separated on precast 4 to 12% bis tris protein gel (Invitrogen) and transferred to a nitrocellulose membrane. Membrane was blocked in PBS–0.05% Tween 20 (PBST) containing 5% nonfat dry milk and incubated overnight at 4˚C with primary antibody diluted in PBST–5% nonfat dry milk or 5% bovine serum albumin (BSA, Sigma). Membranes were subsequently washed with PBST and incubated with the appropriate IRDye 680RD and IRDye 800CW secondary antibody (LI-COR Biosciences) diluted in PBST–5% nonfat dry milk. Images were detected using the Odyssey Systems (LI-COR Biosciences). The following primary antibodies were used: Stat5 (9363T), Phospho-Stat5 (4322T), STAT1 (9172S), Phospho-Stat1 (9167S), JAK2 (3230S), GAPDH (2118S), Hemoglobin-Beta (SC-21757; Santa Cruz). All antibodies were purchased from Cell Signaling (Cell Signaling Technology), unless otherwise noted.

## Supporting information

**S1 Fig. SgRNAs targeting the JAK2 V617 locus. (a)** Schematic indicating positions of all sgRNAs designed and tested.
(TIF)

**S2 Fig. Characterization of JAK2 V617F and 'corrected' F617V HUDEP clones. (a)** DNA gel showing screening of V617F clones by PCR and BsaXI restriction digest. Lower band corresponds to WT allele and higher undigested fragment corresponds to V617F allele. **(b)** Sanger sequencing traces of V617F homozygous (F1) and heterozygous (H1) clones. **(c)** Viability curve depicting cell death in the absence of EPO. F1 and F3 V617F homozygote clones exhibited mildly higher cell viability than WT cells. Data is from n = 5 independent biological replicates. Mean of all experiments ± SD shown. **(d)** Sanger sequencing traces of F617V homozygous (cF1) and heterozygous (cH2) clones. **(e)** Immunoblot and signal intensity

quantification show elevated phosphorylated STAT5 (P-STAT5) and uniform JAK2 expression in JAK2 V617F HUDEP-2 clones without erythropoietin (EPO). Signals were normalized to STAT5 and GAPDH. White bars, JAK2 WT clones; light pink bar, JAK2 V617F heterozygous clone; magenta bars, JAK2 V617F homozygous clones. **(f)** Immunoblot and signal intensity quantification show elevated phosphorylated STAT1 (P-STAT1) expression in JAK2 V617F HUDEP-2 clones both with and without erythropoietin (EPO). Signals were normalized to STAT1 and GAPDH. White bars, JAK2 WT clones; light pink bars, JAK2 V617F heterozygous clones; magenta bars, JAK2 V617F homozygous clones. **(g)** Representative FACS plots for gating live HUDEP-2s expressing Kusabira Orange, a marker gene indicative of viable HUDEP-2s. **(h)** Levels of Glycophorin A (GlyA), an erythroid-specific cell surface marker, in undifferentiated or fully differentiated HUDEP-2s at day 0 and day 5, respectively.
(TIF)

**S3 Fig. Analysis of editing outcomes in subpopulations of HSPCs. (a)** Gating schematic for subsets of CD34+ HSPCs including HSCs (CD34+ CD38- CD45RA- CD90+), multipotent progenitors (MPPs) (CD34+ CD38- CD45RA- CD90-), multipotent lymphoid progenitors (MLPs) (CD34+ CD38- CD45RA+ CD90-/lo), common myeloid progenitors (CMPs) (CD34+ CD38+ CD45RA- CD10- CD135+), megakaryocyte-erythroid progenitors (MEPs) (CD34+ CD38+ CD45RA- CD10- CD135-), and B/NK cells (CD34+ CD38+ CD45RA+ CD10-). **(b)** Composition of HSPC subsets (HSC, MLP, MPP, and CMP) in culture at the time of edit and 3 days post-edit as determined by flow cytometry using gating strategy described in **(a)**. **(c)** HDR-mediated outcomes in HSPC subsets were assessed by amplicon-NGS 3 days after electroporation. Data from n = 3 independent biological replicates. Mean±SD shown. **(d)** NHEJ-mediated outcomes of cells in **(c)** were assessed by amplicon-NGS 3 days after electroporation. Data from n = 3 biological replicates. Mean±SD shown. **(e)** Fraction of CD34+ HSPC subpopulations in V617F or V617V edited CD34+ bulk cells after 4 days of edit. Data from n = 2 independent biological replicates. Mean±SD shown.
(TIF)

**S1 Table. Protospacer and primer sequences. (a)** Protospacer sequences and proximities to target site of guides shown in. **(b)** Sequences of locus-specific primer sets used for T7E1 assay, clonal screening, and amplicon-NGS. **(c)** Sequences of ssODNs used to generate 617V and 617F mutations.
(TIF)

**S1 Raw images.**
(PDF)

## Author Contributions

**Conceptualization:** Ron Baik, Shaheen Kabir, Jacob E. Corn.

**Data curation:** Ron Baik, Stacia K. Wyman, Shaheen Kabir.

**Formal analysis:** Ron Baik, Stacia K. Wyman, Shaheen Kabir.

**Funding acquisition:** Jacob E. Corn.

**Investigation:** Ron Baik, Shaheen Kabir.

**Methodology:** Ron Baik, Shaheen Kabir.

**Supervision:** Shaheen Kabir, Jacob E. Corn.

**Validation:** Ron Baik.

**Visualization:** Ron Baik.

**Writing – original draft:** Ron Baik, Shaheen Kabir, Jacob E. Corn.

**Writing – review & editing:** Ron Baik, Shaheen Kabir, Jacob E. Corn.

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
