## [Decision Letter · Decision Letter 0]

29 Jan 2021

PONE-D-20-40181

Genome editing to model and reverse a prevalent mutation associated with myeloproliferative neoplasms

PLOS ONE

Dear Dr. Corn,

Thank you for submitting your manuscript to PLOS ONE. After careful consideration, we feel that it has merit but does not fully meet PLOS ONE’s publication criteria as it currently stands. Therefore, we invite you to submit a revised version of the manuscript that addresses the points raised during the review process by Reviewer #1.

We look forward to receiving your revised manuscript.

Kind regards,

Francesco Bertolini, MD, PhD

Academic Editor

PLOS ONE

Journal Requirements:

2.) PLOS ONE now requires that authors provide the original uncropped and unadjusted images underlying all blot or gel results reported in a submission’s figures or Supporting Information files. This policy and the journal’s other requirements for blot/gel reporting and figure preparation are described in detail at https://journals.plos.org/plosone/s/figures#loc-blot-and-gel-reporting-requirements and https://journals.plos.org/plosone/s/figures#loc-preparing-figures-from-image-files. When you submit your revised manuscript, please ensure that your figures adhere fully to these guidelines and provide the original underlying images for all blot or gel data reported in your submission. See the following link for instructions on providing the original image data: https://journals.plos.org/plosone/s/figures#loc-original-images-for-blots-and-gels.

3.) Please amend the manuscript submission data (via Edit Submission) to include author Ron Baik; Stacia K. Wyman; Shaheen Kabir.

4.) Thank you for stating the following in your Competing Interests section: 

"NO"

Reviewers' comments:

Reviewer's Responses to Questions

**Comments to the Author**

1. Is the manuscript technically sound, and do the data support the conclusions?

Reviewer #1: Yes

Reviewer #2: Yes

2. Has the statistical analysis been performed appropriately and rigorously? 

Reviewer #1: Yes

Reviewer #2: Yes

3. Have the authors made all data underlying the findings in their manuscript fully available?

Reviewer #1: Yes

Reviewer #2: Yes

4. Is the manuscript presented in an intelligible fashion and written in standard English?

Reviewer #1: Yes

Reviewer #2: Yes

5. Review Comments to the Author

Reviewer #1: The authors reported on an array of experiments using Cas9-based reagents to create and reverse the JAK2 V617F mutation in an immortalized human erythroid progenitor cell line (HUDEP-2), CD34+ adult human hematopoietic stem and progenitor cells, and immunophenotypic long-term hematopoietic stem cells (LT-HSCs).

The first question was whether acquiring the mutation in human hematopoietic cell lineages is sufficient for clonal transformation. They found that : 1. the V617F allele is haplosufficient in hyperactivating JAK2 signalling; 2. the JAK2 V617F homozygous clones exhibited a very modest increase in growth rate over WT clones; 3. acquisition of the JAK2 V617F mutation leads to a competitive growth advantage over WT cells in a mixed ex vivo setting; 4. In the co-culture setting, JAK2V617F was over-represented in the presence of EPO, but the abundance of the JAK2 V617F allele was markedly enhanced in the absence of EPO; 5. There was no difference in differentiation between WT and either heterozygous or homozygous JAK2-V617F clones. They summarized these results by stating that JAK2V617F is sufficient to engender clonal expansion. But they could not claim on the basic question whether JAK2V67F is sufficient to initiate the phenotypically evident disease.

The role of JAK2V617F in clonal myeloproliferative diseases has been the aim of many different studies. Some of them arrived to the conclusion that on its own, JAK2V617F is insufficient to initiate disease (Hasan S, et al. Blood. 2013;122(8):1464-1477; Mullally A, et al. Blood. 2013;121(18):3692-3702; Li J, et al. Blood. 2014;123(20):3139-3151). Thus, the results of this study do not add a great bulk of novelty. However, the use of a minimal system in which the mutation is introduced to the endogenous locus with as few confounders as possible, makes the results worth of consideration.

A major comment is that a synthesis of the results, and which of them is confirmatory of previous evidence and which is new, is lacking, For example the results of phospho- STAT5 should be discussed in the context of very discordant results published in literature.

A further part of the paper was on the effect of V617F mutation on in vitro differentiation. In the results section a long discussion is made about the protocol of editing. This part is difficult to understand by the non experts in the editing methods and we suggest to move it in the methods section.

Reviewer #2: Baik and colleagues presented a genome-editing model to generate JAK2 p.V617F positive human cell lines as well as primary hematopoietic stem and progenitor cells.

They could show that p.V617F cells do not have an increased proliferation rate, but a highly competitive advantage in growth compared with the wild-type in a co-culture system. Furthermore, the Authors found that p.V617F allele is sufficient to increase STAT5 signaling even in the absence of EPO. The strength of these results is that they were not found in an artificial model system like retrovirally transduced murine HSCs. Several studies show a great variety of effects in transgenic mouse models and the development of myeloproliferative neoplasms or acute leukemia. Human genome-edited cells are good complement, or even an alternative to the animal models.

The manuscript show straight-forward hypothesis generating work, and complex result generating technical approaches were used (CRISPR, NGS, flow cytometry, western blotting). In my opinion the reviewer comments (Reviewer #1-3) were carefully and fully answered.

6. PLOS authors have the option to publish the peer review history of their article (what does this mean?). If published, this will include your full peer review and any attached files.

Reviewer #1: No

Reviewer #2: **Yes: **Stephan Bartels

---

## [Author Response · Author response to Decision Letter 0]

5 Feb 2021

1. We thank Reviewer #1 for raising an important point about explaining our results in the context of current literature. As such we have updated our discussion section to more clearly explain what is new and what is confirmatory, as well as including the apt references they suggest. Specifically, please refer to lines 563-565 describing the insufficient role of JAK2 V617F in initiating full-blown MPN and 581-584 for descriptions of JAK2-STAT5 signaling. 

2. We thank the reviewer for suggestions on how to clarify our manuscript. We have moved experimental details of the in vitro differentiation protocol and the list of surface markers used for immunophenotyping from the results section into the methods. Further, we have included a table to display some of the results that were previously described in the text (see Table S1d).

---

## [Decision Letter · Decision Letter 1]

15 Feb 2021

Genome editing to model and reverse a prevalent mutation associated with myeloproliferative neoplasms

PONE-D-20-40181R1

Dear Dr. Corn,

We’re pleased to inform you that your manuscript has been judged scientifically suitable for publication and will be formally accepted for publication once it meets all outstanding technical requirements.

Kind regards,

Francesco Bertolini, MD, PhD

Academic Editor

PLOS ONE

Additional Editor Comments (optional):

Reviewers' comments:

Reviewer's Responses to Questions

**Comments to the Author**

1. If the authors have adequately addressed your comments raised in a previous round of review and you feel that this manuscript is now acceptable for publication, you may indicate that here to bypass the “Comments to the Author” section, enter your conflict of interest statement in the “Confidential to Editor” section, and submit your "Accept" recommendation.

Reviewer #1: All comments have been addressed

2. Is the manuscript technically sound, and do the data support the conclusions?

Reviewer #1: (No Response)

3. Has the statistical analysis been performed appropriately and rigorously? 

Reviewer #1: (No Response)

4. Have the authors made all data underlying the findings in their manuscript fully available?

Reviewer #1: (No Response)

5. Is the manuscript presented in an intelligible fashion and written in standard English?

Reviewer #1: (No Response)

6. Review Comments to the Author

Reviewer #1: (No Response)

7. PLOS authors have the option to publish the peer review history of their article (what does this mean?). If published, this will include your full peer review and any attached files.

Reviewer #1: No

---

## [Editor Report · Acceptance letter]

23 Feb 2021

PONE-D-20-40181R1 

Genome editing to model and reverse a prevalent mutation associated with myeloproliferative neoplasms 

Dear Dr. Corn:

I'm pleased to inform you that your manuscript has been deemed suitable for publication in PLOS ONE. Congratulations! Your manuscript is now with our production department. 

Kind regards, 

on behalf of

Dr. Francesco Bertolini 

Academic Editor

PLOS ONE